# KeyCLD: Learning Constrained Lagrangian Dynamics in Keypoint Coordinates from Images

## Abstract

We present KeyCLD, a framework to learn Lagrangian dynamics from images. Learned keypoint representations derived from images are directly used as positional state vector for jointly learning constrained Lagrangian dynamics. KeyCLD is trained unsupervised end-to-end on sequences of images. Our method explicitly models the mass matrix, potential energy and the input matrix, thus allowing energy based control. We demonstrate learning of Lagrangian dynamics from images on the `dm_control` pendulum, cartpole and acrobot environments, wether they are unactuated, underactuated or fully actuated. Trained models are able to produce long-term video predictions, showing that the dynamics are accurately learned. Our method strongly outperforms recent works on learning Lagrangian or Hamiltonian dynamics from images. The benefits of including a Lagrangian prior and prior knowledge of a constraint function is further investigated and empirically evaluated.

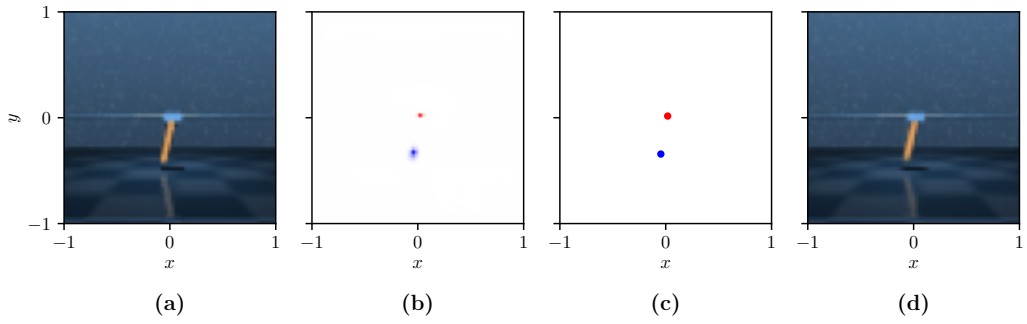

Figure 1: KeyCLD learns Lagrangian dynamics from images. **(a)** An observation of a dynamical system is processed by a learned keypoint estimator model. **(b)** The model represents the positions of the keypoints with a set of spatial probability heatmaps. **(c)** Cartesian coordinates are extracted using spatial softmax and used as positional state vector to learn Lagrangian dynamics. **(d)** The information in the keypoint coordinates bottleneck suffices for a learned renderer model to reconstruct the original observation, including background, reflections and shadows. The keypoint estimator model, Lagrangian dynamics models and renderer model are jointly learned unsupervised on sequences of images.

## 1 Introduction and Related Work

Learning dynamical models from data is a crucial aspect while striving towards intelligent agents interacting with the physical world. Understanding the dynamics and being able to predict future states is paramount for controlling autonomous systems or robots interacting with their environment. For many dynamical systems, the equations of motion can be derived from scalar functions such as the Lagrangian or Hamiltonian. This strong physics prior enables more data-efficient learning and holds energy conserving properties. Greydanus et al. (2019) introduced Hamiltonian neural networks. By using Hamiltonian mechanics as inductive bias, the model respects exact energy conservation laws. Lutter et al. (2018; 2019) pioneered the use of Lagrangian mechanics as physics priors for learning dynamical models from data. Cranmer et al. (2020) expanded this idea to a more general

setting. By modelling the Lagrangian itself with a neural network instead of explicitly modelling mechanical kinetic energy, they can model physical systems beyond classical mechanics. Zhong et al. (2020) included external input forces and energy dissipation, and introduced energy-based control by leveraging the learned energy models. Finzi et al. (2020) introduced learning of Lagrangian or Hamiltonian dynamics in Cartesian coordinates, with explicit constraints. This enables more data efficient models, at the cost of providing extra knowledge about the system in the form of a constraint function.

**Learning Lagrangian dynamics from images**  It is often not possible to observe the full state of a system directly. Cameras provide a rich information source, containing the full state when properly positioned. However, the difficulty lies in interpreting the images and extracting the underlying state. As was recently argued by Lutter & Peters (2021), learning Lagrangian or Hamiltonian dynamics from realistic renderings remains an open challenge. The majority of related work (Greydanus et al., 2019; Toth et al., 2020; Saemundsson et al., 2020; Allen-Blanchette et al., 2020; Botev et al., 2021) use a variational auto-encoder (VAE) framework to represent the state in a latent space embedding. The dynamics model is expressed in this latent space. Zhong & Leonard (2020) use interpretable coordinates, however need full knowledge of the kinematic chain, and the images are segmented per object. Table 1 provides an overview of closely related work in literature.

Table 1: An overview of closely related Lagrangian or Hamiltonian models. Lag-caVAE (Zhong & Leonard, 2020) is capable of modelling external forces and learning from images, but individual moving bodies need to be segmented in the images, on a black background. It additionally needs full knowledge of the kinematic chain, which is more prior information than the constraint function necessary for our method (see Section 2). HGN (Toth et al., 2020) needs no prior knowledge of the kinematic chain, but is unable to model external forces. CHNN (Finzi et al., 2020) expresses Lagrangian or Hamiltonian dynamics in Cartesian coordinates, but can not be learned from images. Our method, KeyCLD, is capable of learning Lagrangian dynamics with external forces, from unsegmented images with shadows, reflections and backgrounds.

|  | HGN | Lag-caVAE | CHNN | KeyCLD |
|---|---|---|---|---|
| External forces (control) |  | ✓ |  | ✓ |
| Interpretable coordinates |  | ✓ | ✓ | ✓ |
| Cartesian coordinates |  |  | ✓ | ✓ |
| Learns from images | ✓ | ✓ |  | ✓ |
| Learns from unsegmented images | ✓ |  |  | ✓ |
| Needs kinematic chain prior |  | ✓ |  |  |
| Needs constraint prior |  |  | ✓ | ✓ |

**Keypoints**  Instead of using VAE inspired latent embeddings, our method leverages fully convolutional keypoint estimator models to observe the state from images. Because the model is fully convolutional, it is also translation equivariant, this leads to a higher data efficiency. Objects can be represented with one or more keypoints, fully capturing the position and orientation. Zhou et al. (2019) used keypoints for object detection, with great success. Keypoint detectors are commonly used for human pose estimation (Zheng et al., 2020). More closely related to this work, keypoints can be learned for control and robotic manipulation (Chen et al., 2021; Vecerik et al., 2021). Minderer et al. (2019) learn unsupervised keypoints from videos to represent objects and dynamics. Jaques et al. (2021) leverage keypoints for system identification and dynamic modelling. Jakab et al. (2018) learn a keypoint representation unsupervised by using it as an information bottleneck for reconstructing images. The keypoints represent semantic landmarks in the images and generalise well to unseen data. It is the main inspiration for the use of keypoints in our work.

**Contributions**  (1) We introduce KeyCLD, a framework to learn constrained Lagrangian dynamics from images. We are the first to use learned keypoint representations from images to learn Lagrangian dynamics. We show that keypoint representations derived from images can directly be used as positional state vector for learning constrained Lagrangian dynamics, expressed in Cartesian coordinates. (2) We show how to control constrained Lagrangian dynamics in Cartesian coordinates with energy shaping, where the state is estimated from images. (3) We adapt the pendulum, cartpole and acrobot

systems from dm_control (Tunyasuvunakool et al., 2020) as benchmarks for learning Lagrangian or Hamiltonian dynamics from images. **(4)** We show that KeyCLD can be learned on these systems, wether they are unactuated, underactuated or fully actuated. We compare quantitatively with Lag-caVAE, Lag-VAE (Zhong & Leonard, 2020) and HGN (Toth et al., 2020), and investigate the benefit of the Lagrangian prior and the constraint function. KeyCLD performs best on all benchmarks.

## 2 Constrained Lagrangian Dynamics

**Lagrangian Dynamics**    For a dynamical system with $m$ degrees of freedom, a set of independent generalized coordinates $\mathbf{q} \in \mathbb{R}^m$ represents all possible kinematic configurations of the system. The time derivatives $\dot{\mathbf{q}} \in \mathbb{R}^m$ are the velocities of the system. If the system is fully deterministic, its dynamics are described by the equations of motion, a set of second order ordinary differential equations (ODE):

$$\ddot{\mathbf{q}} = \mathbf{f}(\mathbf{q}(t), \dot{\mathbf{q}}(t), t, \mathbf{u}(t)) \tag{1}$$

where $\mathbf{u}(t)$ are the external forces acting on the system. From a known initial value $(\mathbf{q}, \dot{\mathbf{q}})$, we can integrate $\mathbf{f}$ through time to predict future states of the system. It is possible to model $\mathbf{f}$ with a neural network, and train the parameters with backpropagation through an ODE solver (Chen et al., 2018).

However, by expressing the dynamics with a Lagrangian we introduce a strong physics prior (Lutter et al., 2019):

$$\mathcal{L}(\mathbf{q}, \dot{\mathbf{q}}) = T(\mathbf{q}, \dot{\mathbf{q}}) - V(\mathbf{q}) \tag{2}$$

$T$ is the kinetic energy and $V$ is the potential energy of the system. For any mechanical system the kinetic energy is defined as:

$$T(\mathbf{q}, \dot{\mathbf{q}}) = \frac{1}{2}\dot{\mathbf{q}}^\top \mathbf{M}(\mathbf{q})\dot{\mathbf{q}} \tag{3}$$

where $\mathbf{M}(\mathbf{q}) \in \mathbb{R}^{m \times m}$ is the positive semi-definite mass matrix. Ensuring that $\mathbf{M}(\mathbf{q})$ is positive semi-definite can be done by expressing $\mathbf{M}(\mathbf{q}) = \mathbf{L}(\mathbf{q})\mathbf{L}(\mathbf{q})^\top$, where $\mathbf{L}(\mathbf{q})$ is a lower triangular matrix. It is now possible to describe the dynamics with two neural networks, one for the mass matrix and one for the potential energy. Since both are only in function of $\mathbf{q}$ and not $\dot{\mathbf{q}}$, and expressing the mass matrix and potential energy is more straightforward than expressing the equations of motion, it is generally much more simple to learn dynamics with this framework. In other words, adding more physics priors in the form of Lagrangian mechanics, makes learning the dynamics more robust and data-efficient (Lutter et al., 2018; 2019; Cranmer et al., 2020; Lutter & Peters, 2021).

The Euler-Lagrange equations (4) allow transforming the Lagrangian into the equations of motion by solving for $\ddot{\mathbf{q}}$:

$$\frac{\mathrm{d}}{\mathrm{dt}}\nabla_{\dot{\mathbf{q}}}\mathcal{L} - \nabla_{\mathbf{q}}\mathcal{L} = \nabla_{\mathbf{q}}W \tag{4}$$

$$\nabla_{\mathbf{q}}W = \mathbf{g}(\mathbf{q})\mathbf{u} \tag{5}$$

where $W$ is the external work done on the system, e.g. forces applied for control. The input matrix $\mathbf{g} \in \mathbb{R}^{m \times l}$ allows introducing external forces $\mathbf{u} \in \mathbb{R}^l$ for modelling any control-affine system. If the external forces and torques are aligned with the degrees of freedom $\mathbf{q}$, $\mathbf{g}$ can be a diagonal matrix or even an identity matrix. More generally, if no prior knowledge is present about the relationship between $\mathbf{u}$ and the generalized coordinates $\mathbf{q}$, $\mathbf{g}(\mathbf{q}) : \mathbb{R}^m \to \mathbb{R}^{m \times l}$ is a function of $\mathbf{q}$ and can be modelled with a third neural network (Zhong et al., 2020). If the system is fully actuated $l = m$, if it is underactuated $l < m$.

**Cartesian coordinates**    Finzi et al. (2020) showed that expressing Lagrangian mechanics in Cartesian coordinates $\mathbf{x} \in \mathbb{R}^k$ instead of independent generalized coordinates $\mathbf{q} \in \mathbb{R}^m$ has several advantages. The mass matrix no longer changes in function of the state, and is thus static. This means that a neural network is no longer required to model the mass matrix, simply the values in the matrix itself are optimized. The potential energy $V(\mathbf{x})$ and input matrix $\mathbf{g}(\mathbf{x})$ are now in function of $\mathbf{x}$. Expressing the potential energy in Cartesian coordinates can often be simpler than in generalized coordinates. E.g. for gravity, this is simply a linear function.

To use the Euler-Lagrange equations without constraint forces, it is required that the system is expressed in independent generalized coordinates, meaning that all possible values of $\mathbf{q}$ correspond to possible states of the system. Since we are now expressing the system in Cartesian coordinates,

$$\Phi(\mathbf{x}) = \begin{bmatrix} \mathbf{e}_y \mathbf{x}_1 - l_1 \\ \|\mathbf{x}_2 - \mathbf{x}_1\|^2 - l_2 \end{bmatrix}$$

Figure 2: Example of a constraint function $\Phi(\mathbf{x})$ to express the cartpole system with Cartesian coordinates. The cartpole system has 2 degrees of freedom, but is expressed in $\mathbf{x} \in \mathbb{R}^4$. Valid configurations of the system in $\mathbb{R}^4$ are constrained on a manifold defined by $\mathbf{0} = \Phi(\mathbf{x})$. The first constraint only allows horizontal movement of $\mathbf{x}_1$, and the second constraint enforces a constant distance between $\mathbf{x}_1$ and $\mathbf{x}_2$. Although unknown $l_1$ and $l_2$ constants are present in $\Phi(\mathbf{x})$, their values are irrelevant, since only the Jacobian of $\Phi(\mathbf{x})$ is used in our framework (see equation (6)). See Appendix A.4 for more examples of constraint functions.

this requirement no longer holds. We need additionally a set of $n$ holonomic constraint functions $\Phi(\mathbf{x}) : \mathbb{R}^k \to \mathbb{R}^n$, where $n$ is the number of constraints so that the degrees of freedom are correct: $m = k - n$. Deriving the equations of motion including the holonomic constraints yields (see Appendix A.1 for the full derivation and details):

$$\mathbf{f} = -\nabla_{\mathbf{x}} V + \mathbf{g}\mathbf{u}$$

$$\ddot{\mathbf{x}} = \mathbf{M}^{-1}\mathbf{f} - \mathbf{M}^{-1}D\Phi^\top \left[ D\Phi\mathbf{M}^{-1}D\Phi^\top \right]^{-1} \left[ D\Phi\mathbf{M}^{-1}\mathbf{f} + \langle D^2\Phi, \dot{\mathbf{x}}\rangle\dot{\mathbf{x}} \right] \tag{6}$$

with $D$ being the Jacobian operator. Since time derivatives of functions modelled with neural networks are no longer present, equation (6) can be implemented in an autograd library which handles the calculation of gradients and Jacobians automatically. See Appendix A.2 for details and the implementation of equation (6) in JAX (Bradbury et al., 2018).

Note that in equation (6) only the Jacobian of $\Phi(\mathbf{x})$ is present. This means that there is no need to learn explicit constants in $\Phi(\mathbf{x})$, such as lengths or distances between points. Rather that constant distances and lengths through time are enforced by $D\Phi(\mathbf{x})\dot{\mathbf{x}} = \mathbf{0}$. We use this property to our advantage since this simplifies the learning process. See Fig. 2 for an example.

The constraint function $\Phi(\mathbf{x})$ adds extra prior information to our model. Alternatively, we could use a mapping function $\mathbf{x} = \mathbf{F}(\mathbf{q})$. This leads directly to an expression of the Lagrangian in Cartesian coordinates using $\dot{\mathbf{x}} = D\mathbf{F}(\mathbf{q})\dot{\mathbf{q}}$:

$$\mathcal{L}(\mathbf{q}, \dot{\mathbf{q}}) = \frac{1}{2}\dot{\mathbf{q}}^\top D\mathbf{F}(\mathbf{q})^\top \mathbf{M} D\mathbf{F}(\mathbf{q})\dot{\mathbf{q}} - V(\mathbf{F}(\mathbf{q})) \tag{7}$$

from which the equations of motion can be derived using the Euler-Lagrange equations, similar to equation (6). In terms of explicit knowledge about the system, the mapping $\mathbf{x} = \mathbf{F}(\mathbf{q})$ is equivalent to the kinematic chain as required for the method of Zhong & Leonard (2020). Using the constraint function is however more general. Some systems, such as systems with closed loop kinematics, can not be expressed in generalized coordinates $\mathbf{q}$, and thus have no mapping function (Betsch, 2005). We therefore argue that adopting the constraint function $\Phi(\mathbf{x})$ is more general and requires less explicit knowledge injected in the model.

**Relationship between Lagrangian and Hamiltonian**    Both Lagrangian and Hamiltonian mechanics ultimately express the dynamics in terms of kinetic and potential energy. The Hamiltonian expresses the total energy of the system $H(\mathbf{q}, \mathbf{p}) = T(\mathbf{q}, \mathbf{p}) + V(\mathbf{q})$ (Greydanus et al., 2019; Toth et al., 2020). It is expressed in the position and the generalized momenta $(\mathbf{q}, \mathbf{p})$, instead of generalized velocities. Using the Legendre transformation it is possible to transform $L$ into $H$ or back. We focus in our work on Lagrangian mechanics because it is more general (Cranmer et al., 2020) and observing the momenta $\mathbf{p}$ is impossible from images. See also Botev et al. (2021) for a short discussion on the differences.

## 3   LEARNING LAGRANGIAN DYNAMICS FROM IMAGES

**Keypoints as state representations**    We introduce the use of keypoints to learn Lagrangian dynamics from images. KeyCLD is trained unsupervised on sequences of $n$ images $\{\mathbf{z}^i\}, i \in \{1, \dots, n\}$ and a constant input vector $\mathbf{u}$. See Fig. 3 for a schematic overview.

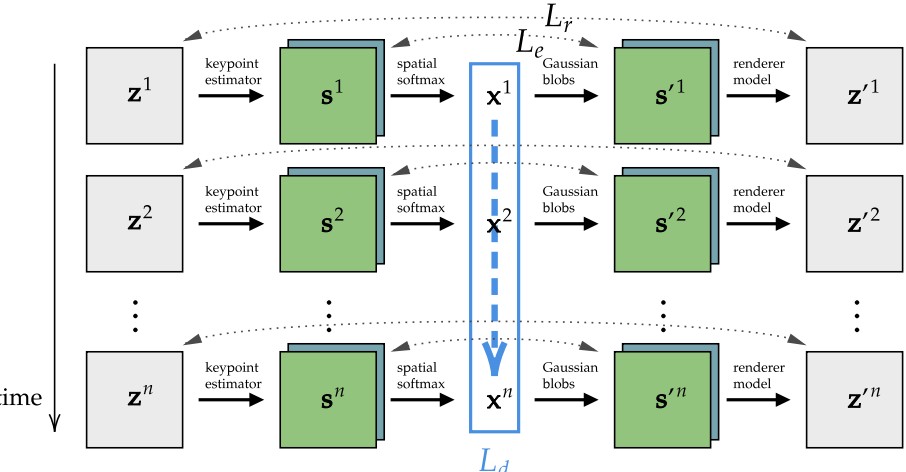

Figure 3: Schematic overview of training KeyCLD. A sequence of $n$ images $\{\mathbf{z}^i\}, i \in \{1, \ldots, n\}$, is processed by the keypoint estimator model, returning heatmaps $\{\mathbf{s}^i\}$ representing spatial probabilities of the keypoints. $\mathbf{s}^i$ consists of $m$ heatmaps $\mathbf{s}_k^i$, one for every keypoint $\mathbf{x}_k^i, k \in \{1, \ldots, m\}$. Spatial softmax is used to extract the Cartesian coordinates of the keypoints, and all keypoints are concatenated in the state vector $\mathbf{x}^i$. $\mathbf{x}^i$ is transformed back to a spatial representation $\mathbf{s}'^i$ using Gaussian blobs. This prior is encouraged on the keypoint estimator model by a binary cross-entropy loss $L_e$ between $\mathbf{s}^i$ and $\mathbf{s}'^i$. The renderer model reconstructs images $\mathbf{z}'^i$ based on $\mathbf{s}'^i$, with reconstruction loss $L_r$. The dynamics loss $L_d$ is calculated on the sequence of state vectors $\mathbf{x}^i$. Keypoint estimator model, renderer model and the dynamics models (mass matrix, potential energy and input matrix) are jointly trained with a weighted sum of the losses $L = L_r + L_e + \lambda L_d$.

All images $\mathbf{z}^i$ in the sequence are processed by the keypoint estimator model, returning each a set of heatmaps $\mathbf{s}^i$ representing the spatial probabilities of keypoint positions. $\mathbf{s}^i$ consists of $m$ heatmaps $\mathbf{s}_k^i, k \in \{1, \ldots, m\}$. The keypoint estimator model is a fully convolutional neural network, maintaining a spatial representation from input to output (see Fig. 4 for the detailed architecture). This contrasts with a model ending in fully connected layers regressing to the coordinates directly, where the spatial representation is lost (Zhong & Leonard, 2020). Because a fully convolutional model is equivariant to translation, it can better generalize to unseen states that are translations of seen states. Another advantage is the possibility of augmenting $\mathbf{z}$ with e.g. random transformations of the $D_4$ dihedral group to increase robustness and data efficiency. Because $\mathbf{s}$ can be transformed back with the inverse transformation, this augmentation is confined to the keypoint estimator model and has no effect on the dynamics.

To distill keypoint coordinates from the heatmaps, we define a Cartesian coordinate system in the image (see for example Fig. 1). Based on this definition, every pixel $\mathbf{p}$ corresponds to a point $\mathbf{x_p}$ in the Cartesian space. The choice of the Cartesian coordinate system is arbitrary but is equal to the space of the dynamics $\ddot{\mathbf{x}}(\dot{\mathbf{x}}, \mathbf{x}, t, \mathbf{u})$ and the constraint function $\mathbf{\Phi}(\mathbf{x})$ (see Section 2). We use spatial softmax over all pixels $\mathbf{p} \in \mathcal{P}$ to distill the coordinates of keypoint $\mathbf{x}_k$ from its probability heatmap:

$$\mathbf{x}_k = \frac{\sum_{\mathbf{p} \in \mathcal{P}} \mathbf{x_p} e^{\mathbf{s}_k(\mathbf{p})}}{\sum_{\mathbf{p} \in \mathcal{P}} e^{\mathbf{s}_k(\mathbf{p})}} \tag{8}$$

Spatial softmax is differentiable, and the loss will backpropagate through the whole heatmap since $\mathbf{x}_k$ depends on all the pixels. Cartesian coordinates $\mathbf{x}_k$ of the different keypoints are concatenated in vector $\mathbf{x}$ which serves as the positional state vector of the system. This compelling connection between image keypoints and Cartesian coordinates forms the basis of this work. The keypoint estimator model serves directly as state estimator to learn Lagrangian dynamics from images.

Similar to Jakab et al. (2018), $\mathbf{x}$ acts as an information bottleneck, through which only the Cartesian coordinates of the keypoints flow to reconstruct the image with the renderer model. First, all $\mathbf{x}_k$ are transformed back to spatial representations $\mathbf{s}'_k$ using Gaussian blobs, parameterized by a

hyperparameter $\sigma$.

$$\mathbf{s}'_k = \exp\left(-\frac{\|\mathbf{x_P} - \mathbf{x}_k\|^2}{2\sigma^2}\right) \tag{9}$$

A binary cross-entropy loss $L_e$ is formulated over $\mathbf{s}$ and $\mathbf{s}'$ to encourage this Gaussian prior. The renderer model can more easily interpret the state in this spatial representation, as it lies closer to its semantic meaning of keypoints as semantic landmarks in the reconstructed image. The renderer model learns a constant feature tensor (inspired by Nguyen-Phuoc et al. (2019)), which provides it with positional information, since the model itself is translation equivariant. See Fig. 4 for the detailed architecture.

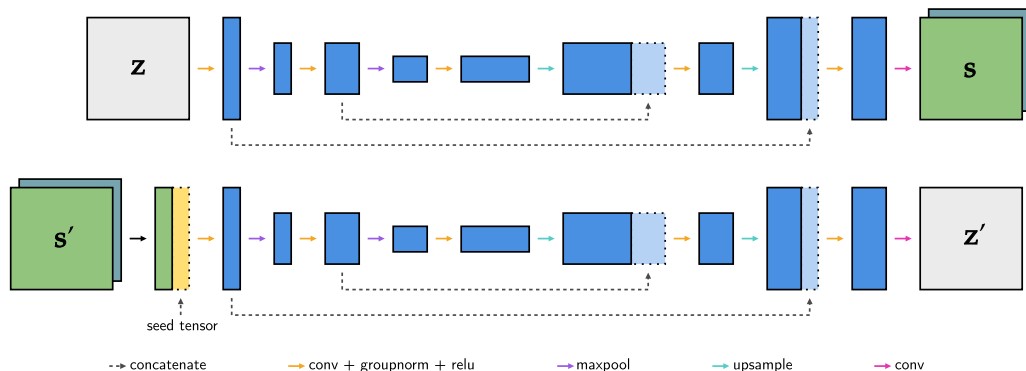

Figure 4: Visualization of the keypoint estimator (top) and renderer (bottom) model architectures. The keypoint estimator model and renderer model have similar architectures, utilizing down- and upsampling and skip connections wich help increasing the receptive field as in Gu et al. (2019); Newell et al. (2016). The renderer model learns a constant feature tensor that is concatenated with the input $\mathbf{s}'$. The feature tensor provides positional information since the fully-convolutional model is translation equivariant.

Finally, a reconstruction loss is formulated over the reconstructed images $\{\mathbf{z}'^i\}$ and original images $\{\mathbf{z}^i\}$:

$$L_r = \sum_{i=1}^{n} \|\mathbf{z}'^i - \mathbf{z}^i\|^2 \tag{10}$$

**Dynamics loss function** The sequence $\{\mathbf{x}^i\}$, corresponding to the sequence of given images $\{\mathbf{z}^i\}$, and the constant input $\mathbf{u}$ is used to calculate the dynamics loss. A fundamental aspect in learning dynamics from images is that velocities can not be directly observed. A single image only captures the position of a system, and contains no information about its velocities[1]. Other work uses sequences of images as input to a model (Toth et al., 2020) or a specific velocity estimator model trained to estimate velocities from a sequence of positions (Jaques et al., 2019). Zhong & Leonard (2020) demonstrate that for estimating velocities, finite differencing simply works better.

We use a central first order finite difference estimation, and project the estimated velocity on the constraints, so that the constraints are not violated:

$$\dot{\mathbf{x}}^i = \left[\boldsymbol{I} - D\boldsymbol{\Phi}(\mathbf{x}^i)^+ D\boldsymbol{\Phi}(\mathbf{x}^i)\right] \frac{\mathbf{x}^{i+1} - \mathbf{x}^{i-1}}{2h} \qquad , i \in \{2, \ldots, n-1\} \tag{11}$$

where $(\cdot)^+$ signifies the Moore-Penrose pseudo-inverse and $h$ the timestep. We can now integrate future timesteps $\hat{\mathbf{x}}$ starting from initial values $(\mathbf{x}^i, \dot{\mathbf{x}}^i)$ using an ODE solver. The equations of motion (6) are solved starting from all initial values in parallel, for $\nu$ timesteps This maximizes the learning signal obtained to learn the dynamics and leads to overlapping subsequences of length $\nu$:

$$\{\hat{\mathbf{x}}^{i+1}, \ldots, \hat{\mathbf{x}}^{i+\nu}\}, \quad i \in \{2, \ldots, n-\nu\} \tag{12}$$

---

[1]Neglecting side-effects such as motion blur, which are not very useful for this purpose.

Thus, $\hat{\mathbf{x}}^{i+j}$ is obtained by integrating $j$ timesteps forward in time, starting from initial value $\mathbf{x}^i$, which was derived by the keypoint estimator model. All $\hat{\mathbf{x}}^{i+j}$ in all subsequences are compared with their corresponding keypoint states $\mathbf{x}^{i+j}$ in an $L_2$ loss:

$$L_d = \sum_{i=2}^{n-\nu} \sum_{j=1}^{\nu} \left\| \mathbf{x}^{i+j} - \hat{\mathbf{x}}^{i+j} \right\|^2 \tag{13}$$

**Total loss**   The total loss is the weighted sum of $L_r$, $L_e$ and $L_d$, with a weighing hyperparameter $\lambda$: $L = L_r + L_e + \lambda L_d$ .

To conclude, the keypoint estimator model, renderer model and dynamics models (mass matrix, potential energy and input matrix) are jointly trained end-to-end on sequences of images $\{\mathbf{z}^i\}$ and constant inputs $\mathbf{u}$ with stochastic gradient descent.

**Rigid bodies as rigid sets of point masses**   By interpreting a set of keypoints as a set of point masses, we can represent any rigid body and its corresponding kinetic and potential energy. Additional constraints are added for the pairwise distances between keypoints representing a single rigid body (Finzi et al., 2020). For 3D systems, at least four keypoints are required to represent any rigid body (Laus & Selig, 2020). We focus in our work on 2D systems in a plane parallel to the camera plane. 2D rigid bodies can be expressed with a set of 2 point masses, which can further be reduced depending on the constraints and connections between bodies. See Appendix A.3 for more details and proof.

## 4 EXPERIMENTS

We adapted the pendulum, cartpole and acrobot environments from `dm_control` (Tunyasuvunakool et al., 2020; Todorov et al., 2012) for our experiments. See Appendix A.4 for details about the environments, their constraint functions and the data generation procedure. The exact same model architectures, hyperparameters and control parameters were used across the environments. This further demonstrates the generality and robustness of our method. See Appendix A.5 for more details.

Since KeyCLD is trained directly on image observations, quantitative metrics can only be expressed in the image domain. The mean square error (MSE) in the image domain is not a good metric of long term prediction accuracy (Minderer et al., 2019; Zhong & Leonard, 2020). A model that trivially learns to predict a static image, which is the average of the dataset, learns no dynamics at all yet this model could report a lower MSE than a model that did learn the dynamics but started drifting from the groundtruth after some time. Therefore, we use the valid prediction time (VPT) score (Botev et al., 2021; Jin et al., 2020) which measures how long the predicted images stay close to the groundtruth images of a sequence:

$$\text{VPT} = \text{argmin}_i [\text{MSE}(\mathbf{z}'^i, \mathbf{z}^i) > \epsilon] \tag{14}$$

where $\mathbf{z}^i$ are the groundtruth images, $\mathbf{z}'^i$ are the predicted images and $\epsilon$ is the error threshold. $\epsilon$ is determined separately for the different environments because it depends on the relative size in pixels of moving parts. We define it as the MSE of the averaged image of the respective validation dataset. Thus it is the lower bound for a model that would simply predict a static image.

We present evaluations with the following ablations and baselines:

| | |
|---|---|
| **KeyCLD** | The full framework as described in Sections 2 and 3. |
| **KeyLD** | The constraint function is omitted. |
| **KeyODE2** | A second order neural ODE modelling the acceleration is used instead of the Lagrangian prior. The keypoint estimator and renderer model are identical to KeyCLD. |
| **Lag-caVAE** | The model presented by Zhong & Leonard (2020). We adapted the model to the higher resolution and color images. |
| **Lag-VAE** | The model presented by Zhong & Leonard (2020). We adapted the model to the higher resolution and color images. |
| **HGN** | Hamiltonian Generative Network presented by Toth et al. (2020). |

Table 2: Valid prediction time (higher is better) in number of predicted frames (mean ± std) for the different models evaluated on the 50 sequences in the validation set. Lag-caVAE and Lag-VAE are only reported on the pendulum environment, since they are unable to model more than one moving body without segmented images. HGN is only reported on non-actuated systems, since it is incapable of modelling external forces and torques. KeyCLD achieves the best results on all benchmarks.

| | # actuators | | **KeyCLD** | KeyLD | KeyODE2 | Lag-caVAE | Lag-VAE | HGN |
|---|---|---|---|---|---|---|---|---|
| Pendulum | 0 | (Fig. 5) | **43.1 ± 9.7** | 16.4 ± 11.3 | 19.1 ± 6.2 | 0.0 ± 0.0 | 10.8 ± 13.8 | 0.2 ± 1.4 |
| | 1 | (Fig. 9) | **39.3 ± 9.8** | 14.9 ± 7.9 | 12.0 ± 4.1 | 0.0 ± 0.1 | 8.0 ± 10.2 | - |
| Cartpole | 0 | (Fig. 10) | **39.9 ± 7.4** | 29.8 ± 11.2 | 29.5 ± 9.5 | - | - | 0.0 ± 0.0 |
| | 1 | (Fig. 11) | **38.4 ± 8.7** | 28.0 ± 9.7 | 24.4 ± 7.9 | - | - | - |
| | 2 | (Fig. 12) | **30.2 ± 10.7** | 23.9 ± 9.6 | 17.7 ± 8.2 | - | - | - |
| Acrobot | 0 | (Fig. 13) | **47.0 ± 6.0** | 40.0 ± 7.9 | 34.3 ± 9.5 | - | - | 2.2 ± 6.9 |
| | 1 | (Fig. 14) | **46.8 ± 4.6** | 29.5 ± 6.3 | 33.0 ± 7.4 | - | - | - |
| | 2 | (Fig. 15) | **47.0 ± 3.5** | 39.1 ± 9.9 | 30.8 ± 9.3 | - | - | - |

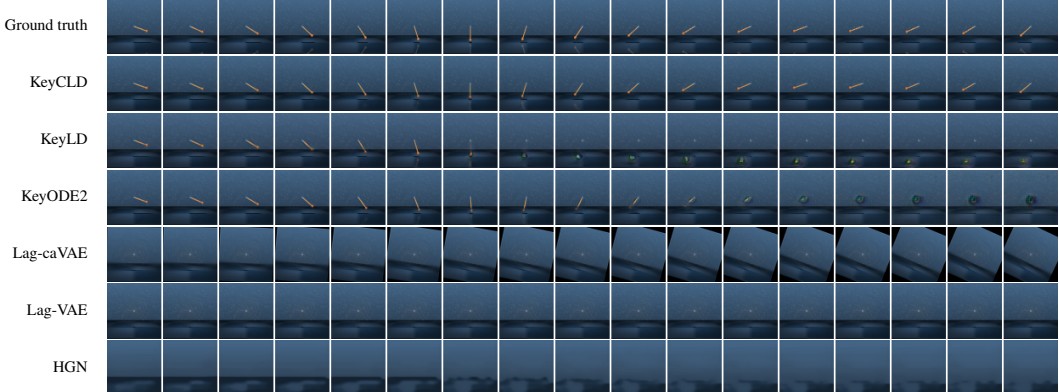

Figure 5: Future frame predictions of the unactuated pendulum. These correspond to the first row in Table 2. 50 frames are predicted based on the first three frames of the ground truth sequence to estimate the velocity. Every third frame of every sequence is shown. KeyCLD is capable of making accurate long-term predictions with minimal drift of the dynamics. Without constraint function, KeyLD is not capable of making long-term predictions. Similarly, KeyODE2 is unable of making long-term predictions. Lag-caVAE is fundamentally incapable of modelling data with background information, since the reconstructed images are explicitly rotated. Lag-VAE does not succeed in modelling moving parts in the data, and simply learns to predict static images. HGN also does not capture the dynamics and only learns the background.

See Table 2 for an overview of results, and Fig. 5 for qualitative results on the unactuated pendulum. KeyCLD achieves the best results on all benchmarks. Lag-caVAE is unable to model data with background (see also Fig. 5). Despite our best efforts for implementation and training, Lag-VAE and HGN perform very poorly. The models are not capable of handling the relatively more challenging visual structure of dm_control environments. Removing the constraint function (KeyLD) has a detrimental effect on the ability to make long-term predictions. Results are comparable to removing the Lagrangian prior altogether (KeyODE2). This suggests that modeling dynamics in Cartesian coordinates coupled with keypoint representations is in itself a very strong prior, consistent with recent findings by Gruver et al. (2021). However, using a Lagrangian formulation allows leveraging a constraint function, since a general neural ODE model can not make use of a constraint function. Thus, if a constraint function is available, the Lagrangian prior becomes much more powerful. See Appendix A.7 for more qualitative results and insights.

**Interpretable energy models and control**    A major argument in favor of expressing dynamics in terms of a mass matrix and potential energy is the straightforward control design via passivity based control and energy shaping. See Appendix A.6 for details and derivation of an energy shaping controller, and successful swing-up results of the pendulum, cartpole and acrobot system.

# 5   Conclusion and Future Work

We introduce the use of keypoints to learn Lagrangian dynamics from images. Learned keypoint representations derived from images are directly used as positional state vector for learning constrained Lagrangian dynamics. The pendulum, cartpole and acrobot systems of `dm_control` are adapted as benchmarks. Previous works in literature on learning Lagrangian or Hamiltonian dynamics from images were benchmarked on very simple renderings of flat sprites on blank backgrounds (Botev et al., 2021; Zhong & Leonard, 2020; Toth et al., 2020), whereas `dm_control` is rendered with lighting effects, shadows, reflections and backgrounds. We believe that working towards more realistic datasets is crucial for applying Lagrangian models in the real world. The challenge of learning Lagrangian dynamics from more complex images should not be underestimated. Despite our best efforts in implementing and training Lag-caVAE, Lag-VAE (Zhong & Leonard, 2020) and HGN (Toth et al., 2020), they perform very poorly on our `dm_control` benchmark. KeyCLD is capable of making long-term predictions and learning accurate energy models, suitable for energy shaping control. When no constraint prior is available, results are comparable to a general second order neural ODE. This signifies the benefit of using keypoint representations coupled with Cartesian coordinates to model the dynamics.

Our work focusses on 2D systems, where the plane of the system is parallel with the camera plane. Elevation to 3D, e.g. setups with multiple cameras, is an interesting future direction. Secondly, modelling contacts by using inequality constraints could be a useful addition. Thirdly, our work focusses on energy-conserving systems. Modelling energy dissipation is necessary for real-world applications. Several recent papers have proposed methods to incorporate energy dissipation in the Lagrangian dynamics models (Zhong et al., 2020; Greydanus & Sosanya, 2022). However, Gruver et al. (2021) argue that modelling the acceleration directly with a second order differential equation and expressing the system in Cartesian coordinates, is a better approach. Further research into both approaches would clarify the benefit of Lagrangian and Hamiltonian priors on real-world applications. Lastly, a major argument in using the Lagrangian prior is the availability of the mass matrix, potential energy and input matrix. This allows explainability and powerful control designs, such as energy shaping. Model-based research for underactuated systems uses stochastic optimal control which often fails since a long prediction horizon is required. With our method, feedback controllers based on the mass matrix, potential energy and input matrix are possible which are more robust and do not require complicated optimal control.

# 6   Broader impact

A tenacious divide exists between control engineering researchers and computer science researchers working on control. Where the first would use known equations of motion for a specific class of systems and investigate system identification, the latter would strive for the most general method with no prior knowledge. We believe this is a spectrum worth exploring, and as such use strong physics priors as Lagrangian mechanics, but still model e.g. the input matrix and the potential energy with arbitrary neural networks. The broad field of model-based reinforcement learning could benefit from decades of theory and practice in classic control theory and system identification. We hope this paper could help bridge both worlds.

Using images as input is, in a broad sense, very powerful. Since camera sensors are consistently becoming cheaper and more powerful due to advancements in technology and scaling opportunities, we can leverage these rich information sources for a deeper understanding of the world our intelligent agents are acting in. Image sensors can replace and enhance multiple other sensor modalities, at a lower cost. This work demonstrates the ability to efficiently model and control dynamical systems that are captured by cameras, with no supervision and minimal prior knowledge. We want to stress that we have shown it is possible to learn both the Lagrangian dynamics and state estimator model from images in one end-to-end process. The complex interplay between both, often makes them the most labour intensive parts in system identification. We believe this is a gateway step in achieving reliable end-to-end learned control from pixels, especially since the availability of mass matrix, potential energy and input matrix models allows powerful control designs.

## REPRODUCIBILITY STATEMENT

Please see the attached codebase to reproduce all experiments reported in this paper. The `README.md` file contains detailed installation instructions and scripts for every experiment.

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

# A  APPENDIX

## A.1  DERIVATION OF CONSTRAINED EULER-LAGRANGE EQUATIONS

The Lagrangian of a mechanical system described in Cartesian coordinates $\mathbf{x} \in \mathbb{R}^k$ is:

$$\mathcal{L}(\mathbf{x}, \dot{\mathbf{x}}) = \frac{1}{2}\dot{\mathbf{x}}^\top \mathbf{M}\dot{\mathbf{x}} - V(\mathbf{x}) \tag{15}$$

with $\mathbf{M}$ a *static* mass matrix, not depending on $\mathbf{x}$, and $V(\mathbf{x})$ the potential energy. If the system has $m$ degrees of freedom, additionally $n$ holonomic constraints are necessary such that $m = k - n$. These are described by a constraint function $\mathbf{\Phi}(\mathbf{x}) : \mathbb{R}^k \to \mathbb{R}^n$. Including the input matrix $\mathbf{g}(\mathbf{x}) \in \mathbb{R}^{k \times l}$ and external inputs $\mathbf{u}(t) \in \mathbb{R}^l$, the constrained Euler-Lagrange equations is expressed with a vector $\boldsymbol{\lambda}(t) \in \mathbb{R}^n$ containing Lagrange multipliers for the constraints (Finzi et al., 2020; Lanczos, 2020):

$$\frac{\mathrm{d}}{\mathrm{dt}}\nabla_{\dot{\mathbf{x}}}\mathcal{L}(\mathbf{x}, \dot{\mathbf{x}}) - \nabla_{\mathbf{x}}\mathcal{L}(\mathbf{x}, \dot{\mathbf{x}}) = \mathbf{g}(\mathbf{x})\mathbf{u}(t) + D\mathbf{\Phi}(\mathbf{x})^\top \lambda(t) \tag{16}$$

Because the mass matrix is static [2] , this is simplified to:

$$\mathbf{M}\ddot{\mathbf{x}} + \nabla_{\mathbf{x}}V(\mathbf{x}) = \mathbf{g}(\mathbf{x})\mathbf{u}(t) + D\mathbf{\Phi}(\mathbf{x})^\top \boldsymbol{\lambda}(t) \tag{17}$$

$$\ddot{\mathbf{x}} = \mathbf{M}^{-1}\mathbf{f} + \mathbf{M}^{-1}D\mathbf{\Phi}(\mathbf{x})^\top \boldsymbol{\lambda}(t), \quad \mathbf{f} = -\nabla_{\mathbf{x}}V(\mathbf{x}) + \mathbf{g}(\mathbf{x})\mathbf{u}(t) \tag{18}$$

Calculating twice the time derivative of the constraint conditions yields:

$$\begin{aligned}
\mathbf{0} &\equiv \mathbf{\Phi}(\mathbf{x}) \\
\mathbf{0} &= \dot{\mathbf{\Phi}}(\mathbf{x}) \\
\mathbf{0} &= D\mathbf{\Phi}(\mathbf{x})\dot{\mathbf{x}} \\
\mathbf{0} &= D\dot{\mathbf{\Phi}}(\mathbf{x})\dot{\mathbf{x}} + D\mathbf{\Phi}(\mathbf{x})\ddot{\mathbf{x}}
\end{aligned} \tag{19}$$

The Lagrange multipliers $\boldsymbol{\lambda}(t)$ are solved by substituting $\ddot{\mathbf{x}}$ from equation (18) in equation (19):

$$\begin{aligned}
-D\dot{\mathbf{\Phi}}(\mathbf{x})\dot{\mathbf{x}} &= D\mathbf{\Phi}(\mathbf{x})\mathbf{M}^{-1}\mathbf{f} + D\mathbf{\Phi}(\mathbf{x})\mathbf{M}^{-1}D\mathbf{\Phi}(\mathbf{x})^\top \boldsymbol{\lambda}(t) \\
\boldsymbol{\lambda}(t) &= \left[D\mathbf{\Phi}(\mathbf{x})\mathbf{M}^{-1}D\mathbf{\Phi}(\mathbf{x})^\top\right]^{-1}\left[D\mathbf{\Phi}(\mathbf{x})\mathbf{M}^{-1}\mathbf{f} + D\dot{\mathbf{\Phi}}(\mathbf{x})\dot{\mathbf{x}}\right]
\end{aligned} \tag{20}$$

We use the chain rule a second time to get rid of the time derivative of $D\mathbf{\Phi}(\mathbf{x})$:

$$D\dot{\mathbf{\Phi}}(\mathbf{x})\dot{\mathbf{x}} = \langle D^2\mathbf{\Phi}, \dot{\mathbf{x}}\rangle\dot{\mathbf{x}} \tag{21}$$

Substituting $\boldsymbol{\lambda}(t)$ in (18) we arrive at equation (6).

## A.2  IMPLEMENTATION OF CONSTRAINED EULER-LAGRANGE EQUATIONS IN JAX

It could seem a daunting task to implement the derivation of the constrained Euler-Lagrange equations (6) in an autograd library. Therefore, we provide an implementation in JAX (Bradbury et al., 2018). For more context, please see the full code base in the supplementary materials (`keycld/models.py`).

```python
import jax
import jax.numpy as jnp

def constraint_fn(x):
    # function that returns a vector with constraint values
    c = jnp.array([
        ...,
    ])
    return c

def mass_matrix(params, x):
    # function that returns the mass matrix
    ...
    return m
```

---

[2]In other words, the centrifugal and Coriolis forces are zero because $\dot{\mathbf{M}} = 0$ and $\nabla_{\mathbf{x}}\mathbf{M} = 0$.

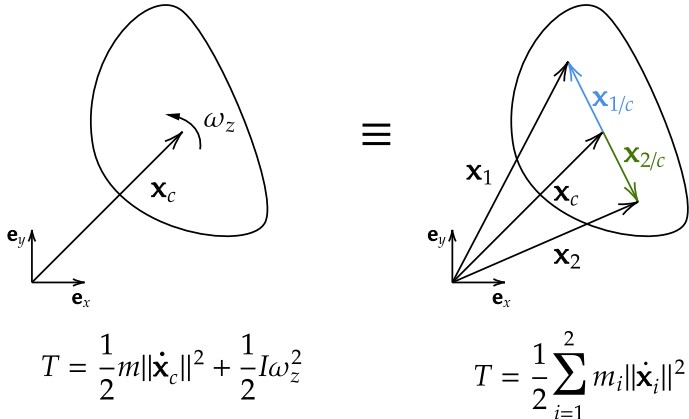

$$T = \frac{1}{2}m\|\dot{\mathbf{x}}_c\|^2 + \frac{1}{2}I\omega_z^2 \qquad\qquad T = \frac{1}{2}\sum_{i=1}^{2} m_i\|\dot{\mathbf{x}}_i\|^2$$

Figure 6: Any 2D rigid body with mass $m$ and rotational inertia $I$ is equivalent to a set of two point masses $\mathbf{x}_1$ and $\mathbf{x}_2$ with masses $m_1$ and $m_2$. The kinetic energy of the rigid body, expressed in a translational part and a rotational part, is equal to the sum of the kinetic energies of the point masses.

```python
def potential_energy(params, x):
    # function that returns the potential energy
    ...
    return V

def input_matrix(params, x):
    # function that returns the input matrix
    ...
    return g

def euler_lagrange(params, x, x_t, action):
    m_inv = jnp.linalg.pinv(mass_matrix(params, x))
    f = - jax.grad(potential_energy, 1)(params, x) + input_matrix(params, x) @ action

    Dphi = jax.jacobian(constraint_fn)(x)
    DDphi = jax.jacobian(jax.jacobian(constraint_fn))(x)

    # Lagrange multiplicators:
    l = jnp.linalg.pinv(Dphi @ m_inv @ Dphi.T) @ (Dphi @ m_inv @ f + DDphi @ x_t @ x_t)
    x_tt = m_inv @ (f - Dphi.T @ l)

    return x_tt
```

## A.3    Rigid bodies as sets of point masses

The position of a rigid body in 2D is fully described by the position of its center of mass $\mathbf{x}_c$ and orientation $\theta$. Potential energy only depends on the position, thus if we want to describe the potential energy with an equivalent rigid set of point masses, two points are sufficient to fully determine $\mathbf{x}_c$ and $\theta$. For the kinetic energy, we provide the following Theorem and proof:

**Theorem 1.** *For any 2D rigid body, described by its center of mass* c*, mass* $m$ *and rotational inertia* I*, there exists an equivalent rigid set of two point masses* $\mathbf{x}_1$ *and* $\mathbf{x}_2$ *with masses* $m_1$ *and* $m_2$.

*Proof.* To find conditions such that the kinetic energy expressed in two point masses should be equal to the rigid body representation, we start by expressing general 3D-movement:

$$\mathbf{x}_i = \mathbf{x}_c + \mathbf{x}_{i/c} \qquad , i \in \{1, 2\} \tag{22}$$

Where the vector $\mathbf{x}_c$ are the coordinates of the center of mass and the vector $\mathbf{x}_{i/c}$ is the position of the point mass relative to the center of mass. Since this relative position $\mathbf{x}_{i/c}$ has fixed length, only a rotation is possible and hence the equation of the velocity is:

$$\dot{\mathbf{x}}_i = \dot{\mathbf{x}}_c + \boldsymbol{\omega} \times \mathbf{x}_{i/c} \qquad , i \in \{1, 2\} \tag{23}$$

where $\boldsymbol{\omega}$ is the rotational velocity of the body. Substituting this in the kinetic energy of the point masses, we get:

$$
\begin{aligned}
T &= \frac{1}{2}\sum_{i=1}^{2} m_i \|\dot{\mathbf{x}}_c + \boldsymbol{\omega} \times \mathbf{x}_{i/c}\|^2 \\
&= \frac{1}{2}\sum_{i=1}^{2} m_i \Big( \|\dot{\mathbf{x}}_c\|^2 + \|\boldsymbol{\omega} \times \mathbf{x}_{i/c}\|^2 + 2\mathbf{x}_{i/c} \cdot \big( \dot{\mathbf{x}}_c \times \boldsymbol{\omega} \big) \Big)
\end{aligned}
\tag{24}
$$

Where we calculated the square and used the circular shift property of the triple product on the last term.

For movement in the 2D-plane (i.e. $\boldsymbol{\omega} = \vec{e}_z \omega_z$ and $\mathbf{x}_i = \vec{e}_x \mathbf{x}_{i,x} + \vec{e}_y \mathbf{x}_{i,y}$), this becomes:

$$
\begin{aligned}
T &= \frac{1}{2}\sum_{i=1}^{2} m_i \Big( \|\dot{\mathbf{x}}_c\|^2 + \|\mathbf{x}_{i/c}\|^2 \omega_z^2 + 2\mathbf{x}_{i/c} \cdot \big( \dot{\mathbf{x}}_c \times \boldsymbol{\omega} \big) \Big) \\
&= \frac{1}{2}(m_1 + m_2)\|\dot{\mathbf{x}}_c\|^2 + \frac{1}{2}\big( m_1 \|\mathbf{x}_{1/c}\|^2 + m_2 \|\mathbf{x}_{2/c}\|^2 \big)\omega_z^2 + \big( m_1 \mathbf{x}_{1/c} + m_2 \mathbf{x}_{2/c} \big) \cdot \big( \dot{\mathbf{x}}_c \times \boldsymbol{\omega} \big)
\end{aligned}
\tag{25}
$$

Matching the kinetic energy of the 2 point masses (equation (25)) with that of the rigid body representation (left hand side of Figure 6), we get following conditions:

$$
\begin{cases}
m = m_1 + m_2 \\
I = m_1 \|\mathbf{x}_{1/c}\|^2 + m_2 \|\mathbf{x}_{2/c}\|^2 \\
\mathbf{0} = m_1 \mathbf{x}_{1/c} + m_2 \mathbf{x}_{2/c}
\end{cases}
\tag{26}
$$

Since the last equation is a vector equation, this gives us four equations in six unknowns $(m_1, m_2, \mathbf{x}_{1,x}, \mathbf{x}_{1,y}, \mathbf{x}_{2,x}, \mathbf{x}_{2,y})$, which leaves us the freedom to choose two. $\qquad\square$

It follows from the third condition of (26) that points $\mathbf{x}_1$, $\mathbf{x}_2$ and $\mathbf{x}_c$ should be collinear. To conclude, we can freely choose the positions of the point masses (as long as $\mathbf{x}_c$ is on the line between them), and will be able to model the rigid body as a set of two point masses. In practice, KeyCLD will freely choose the keypoint positions to be able to model the dynamics. Depending on the constraints in the system, it is possible to further reduce the number of necessary keypoints. See Appendix A.4 for examples.

The interpretation of rigid bodies as sets of point masses allows expressing the kinetic energy as the sum of the kinetic energies of the point masses. The mass matrix for a 2D system is thus defined as a diagonal matrix with masses $m_k$ for every keypoint $\mathbf{x}_k$, leading to the following expression for the kinetic energy of the system:

$$
T(\dot{\mathbf{x}}) = \frac{1}{2}\dot{\mathbf{x}}^\top \mathbf{M}\dot{\mathbf{x}} = \frac{1}{2}\begin{bmatrix} \dot{\mathbf{x}}_1 & \dots & \dot{\mathbf{x}}_n \end{bmatrix}\begin{bmatrix} m_1 & 0 & \dots & 0 & 0 \\ 0 & m_1 & \dots & 0 & 0 \\ \vdots & \vdots & \ddots & \vdots & \vdots \\ 0 & 0 & \dots & m_n & 0 \\ 0 & 0 & \dots & 0 & m_n \end{bmatrix}\begin{bmatrix} \dot{\mathbf{x}}_1 \\ \vdots \\ \dot{\mathbf{x}}_n \end{bmatrix}
\tag{27}
$$

To enforce positive values, the masses are parameterized by their square root and squared.

## A.4   Details about the DM_CONTROL environments and data generation

We adapted the pendulum, cartpole and acrobot environments from dm_control (Tunyasuvunakool et al., 2020) implemented in MuJoCo (Todorov et al., 2012). Both are released under the Apache-2.0 license. Following changes were made to the environments to adapt them to our use-case:

**Pendulum**   The camera was repositioned so that it is in a parallel plane to the system. Friction was removed. Torque limits of the motor are increased.

**Cartpole**   The camera was moved further away from the system to enable a wider view, the two rails are made longer and the floor lowered so that they are not cut-off with the wider view. All friction is removed. The pole is made twice as thick, the color of the cart is changed. Torque limits are increased and actuation is added to the cart to make full actuation possible.

**Acrobot**   The camera and system are moved a little bit upwards. The two poles are made twice as thick, and one is changed in color. Torque limits are increased and actuation is added to the upper part to make full actuation possible.

**Data generation**   For every environment, 500 runs of 50 timesteps are generated with a 10% validation split. Initial state for every sequence is at random position with small random velocity. The control inputs $\mathbf{u}$ are constant throughout a sequence, and uniform randomly chosen between the force and torque limits of the input. We set $\mathbf{u} = \mathbf{0}$ for 20% of the sequences. We found this helps the model to learn the dynamics better, discouraging confusion of the energy models with external actions.

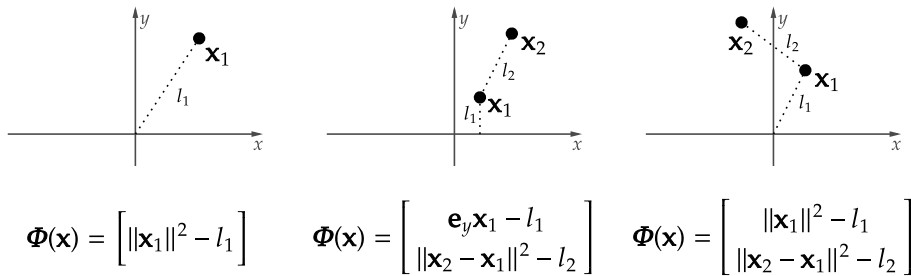

$$\boldsymbol{\Phi}(\mathbf{x}) = \begin{bmatrix} \|\mathbf{x}_1\|^2 - l_1 \end{bmatrix} \qquad \boldsymbol{\Phi}(\mathbf{x}) = \begin{bmatrix} \mathbf{e}_y \mathbf{x}_1 - l_1 \\ \|\mathbf{x}_2 - \mathbf{x}_1\|^2 - l_2 \end{bmatrix} \qquad \boldsymbol{\Phi}(\mathbf{x}) = \begin{bmatrix} \|\mathbf{x}_1\|^2 - l_1 \\ \|\mathbf{x}_2 - \mathbf{x}_1\|^2 - l_2 \end{bmatrix}$$

Figure 7: From left to right the pendulum, cartpole and acrobot `dm_control` environments. The respective constraint functions are given below each schematic.

The constraint function for each of the environments are given in Fig. 7. As explained in Appendix A.3, every rigid body needs to be represented by two keypoints. But due to the constraints it is possible to omit certain keypoints, because they do not move or coincide with other keypoints. As experimentally validated, we can thus model all three systems with a lower number of keypoints, where the number of keypoints equals the number of bodies.

**Pendulum**   One keypoint is used to model the pendulum. The second keypoint of this rigid body can be omitted because it can be assumed to be at the origin. Due to the constraint function, this point will provide no kinetic energy since it will not move. Since the other keypoints position and mass is freely chosen, any pendulum can be modelled. The constraint function expresses that the distance $l_1$ from the origin to $\mathbf{x_1}$ is fixed. The value of $l_1$ in the implementation is irrelevant because it vanishes when taking the Jacobian.

**Cartpole**   Two keypoints are used to model the cartpole. The constraint function expresses that $\mathbf{x}_1$ does not move in the vertical direction and the distance $l_1$ between $\mathbf{x}_1$ and $\mathbf{x}_2$ is constant. Again, the values of $l_1$ and $l_2$ in the implementation are irrelevant.

**Acrobot**   Two keypoints are used to model the acrobot. The constraint function expresses that lengths $l_1$ and $l_2$ are constant through time. Again, the values are irrelevant in the implementation.

A.5   TRAINING HYPERPARAMETERS AND DETAILS

All models were trained on one NVIDIA RTX 2080 Ti GPU.

**KeyCLD, KeyLD and KeyODE2**   We use the Adam optimizer (Kingma & Ba, 2015), implemented in Optax (Hessel et al., 2020) with a learning rate of $3 \times 10^{-4}$. We use the exact same hyperparameters for all the environments and did not tune them individually. Dynamics loss weight $\lambda = 1$, $\sigma = 0.1$

for the Gaussian blobs in $\mathbf{s}'$. The hidden layers in the keypoint estimator and renderer model have at the first block 32 features, this increases to respectively 64 and 128 after every maxpool operation. All convolutions have kernel size $3 \times 3$, and maxpool operations scale down with factor 2 with a kernel size of $2 \times 2$.

The potential energy is modelled with an MLP with two hidden layers with 32 neurons and celu activation functions (Barron, 2017). The weights are initialized with a normal distribution with standard deviation 0.01. Likewise, the input matrix is modelled with an MLP similar to the potential energy. The ouputs of this MLP are reshaped in the required shape of the input matrix.

The KeyODE2 dynamics model is an MLP with three hidden layers with each 64 neurons. We chose a higher number of layers and neurons, to allow this model more expressivity compared to the potential energy and input matrix models of KeyCLD.

**Lag-caVAE, Lag-VAE and HGN**  For the Lag-caVAE and Lag-VAE baselines, the official public codebase was used (Zhong & Leonard, 2020). We adapted the implementation to work with the higher input resolution of 64 by 64 (instead of 32 by 32), and 3 color channels (instead of 1).

For the HGN baseline, we used the implementation that was also released by Zhong & Leonard (2020). The architecture was adapted to work with the higher input resolution of 64 by 64 (instead of 32 by 32) by adding an extra upscale layer in the decoder, and a maxpool layer and one extra convolutional layer in the encoder.

## A.6 ENERGY SHAPING CONTROL

A major argument in favor of expressing dynamics in terms of a mass matrix and potential energy is the straightforward control design via passivity based control and energy shaping (Ortega et al., 2001). Recent works of Zhong et al. (2020); Zhong & Leonard (2020) use energy shaping in generalized coordinates. In Cartesian coordinates, energy shaping can still be used. This is easily seen from the fact that for the holonomic constraints $\mathbf{\Phi}(\mathbf{x}) \equiv 0$, we have the derivative $D\mathbf{\Phi}(\mathbf{x})\dot{\mathbf{x}} = \mathbf{0}$, which means that the constraint forces in equation (6) are perpendicular to the path and hence do no work nor influence the energy (Lanczos, 2020).

Energy shaping control makes sure that the controlled system behaves according to a potential energy $V_d(\mathbf{x})$ instead of $V(\mathbf{x})$:

$$\mathbf{u} = (\mathbf{g}^\top \mathbf{g})^{-1} \mathbf{g}^\top \left( \nabla_{\mathbf{x}} V - \nabla_{\mathbf{x}} V_d \right) - y_{\text{passive}} \tag{28}$$

where $y_{\text{passive}}$ can be any passive output, the easiest choice being $y_{\text{passive}} = k_d \mathbf{g}^\top \dot{\mathbf{x}}$, where $k_d$ is a tuneable control parameter. The proposed potential energy $V_d$ should be such that:

$$\begin{aligned} \mathbf{x}^* &= \text{argmin} V_d(\mathbf{x}) \\ \mathbf{0} &= \mathbf{g}^\perp \left( \nabla_{\mathbf{x}} V - \nabla_{\mathbf{x}} V_d \right) \end{aligned} \tag{29}$$

Where $\mathbf{g}^\perp$ is the left-annihilator of $\mathbf{g}$, meaning that $\mathbf{g}^\perp \mathbf{g} = \mathbf{0}$. For fully actuated systems, the first condition of equation (29) is always met and the easiest choice is:

$$V_d(\mathbf{x}) = (\mathbf{x} - \mathbf{x}^*)^\top k_p (\mathbf{x} - \mathbf{x}^*) \tag{30}$$

where $k_p$ is a tuneable control parameter. The desired equilibrium position $\mathbf{x}^*$ is obtained by putting a picture of the desired position of the system into the keypoint estimator model. Finally, the passivity-based controller that is used is:

$$\mathbf{u} = (\mathbf{g}^\top \mathbf{g})^{-1} \mathbf{g}^\top \left[ \nabla_{\mathbf{x}} V - k_p (\mathbf{x} - \mathbf{x}^*) \right] - k_d \mathbf{g}^\top \dot{\mathbf{x}} \tag{31}$$

Changing the behavior of the kinetic energy is also possible (Gomez-Estern et al., 2001), but if left for future work. The passivity-based controller is (see Appendix A.6 for full derivation and details):

$$\mathbf{u} = (\mathbf{g}^\top \mathbf{g})^{-1} \mathbf{g}^\top \left[ \nabla_{\mathbf{x}} V - k_p (\mathbf{x} - \mathbf{x}^*) \right] - k_d \mathbf{g}^\top \dot{\mathbf{x}} \tag{32}$$

Many model-based reinforcement learning algorithms require the learning of a full neural network as controller. Whilst in this work, due to knowledge of the potential energy, we only need to tune two parameters $k_p$ and $k_d$.

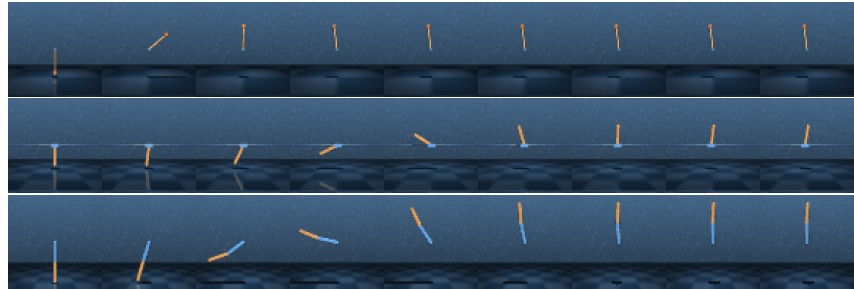

Figure 8: KeyCLD allows using energy shaping control because the learned potential energy model is available. Based on a swing-up target image $\mathbf{z}^*$, the target state $\mathbf{x}^*$ is determined by the keypoint detector model. The sequences show that all three systems can achieve the target state. The control parameters $k_p = 5.0$ and $k_d = 2.0$ are the same for all systems, demonstrating the generality of the control method.

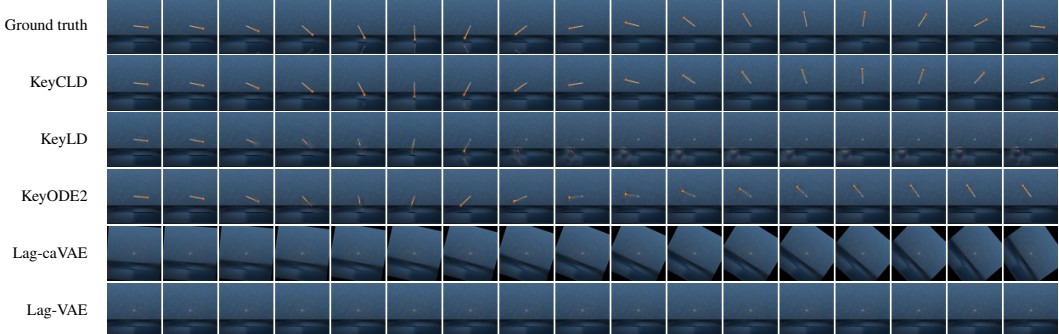

Figure 9: Actuated pendulum.

See Figure 8 for results of successful swing-up of the pendulum, cartpole and acrobot system. The same control parameters $k_p = 5.0$ and $k_d = 2.0$ are used for all systems, demonstrating the generality of the control method.

## A.7 ADDITIONAL EXPERIMENTAL RESULTS

Here we present additional qualitative results. Please refer to the supplementary materials for movies.

**Future frame predictions** We generate predictions of 50 frames, given the first 3 frames of the ground truth sequence to estimate the initial velocity. Please compare these qualitative results for the unactuated and actuated pendulum environment(Fig. 5, 9), unactuated, underactuated and fully actuated cartpole environment (Fig. 10, 11, 12) and unactuated, underactuated and fully actuated acrobot environment (Fig. 13, 14, 15). Every third frame of the sequence is shown. See also the supplementary materials for movies of all sequences in the validation set. For every environment very long predictions of 500 frames are included, including visualizations of the keypoint representations and predictions.

**Learned potential energy models** Since the potential energy $V$ is explicitly modelled, we can plot values throughout sequences of the state space. A sequence of images is processed by the learned keypoint estimator model, and the states are then used to calculate the potential energy with the learned potential energy model. Absolute values of the potential energy are irrelevant, since the potential is relative, but we gain insights by moving parts of the system separately. See Figure 16 for results for the pendulum, Figures 17 and 18 for the cartpole and Figures 19 and 20 for the acrobot.

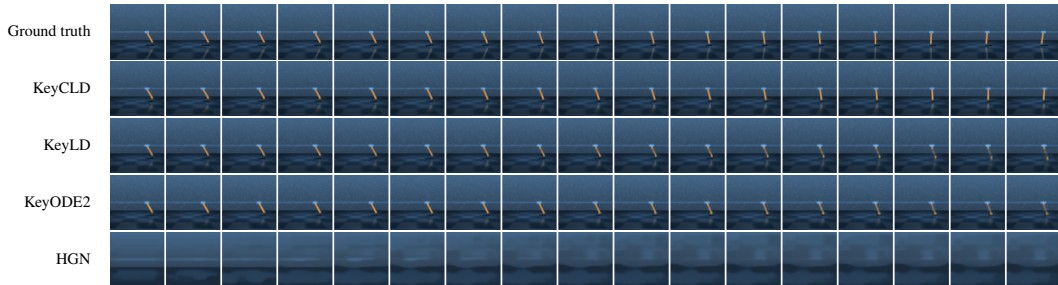

Figure 10: Unactuated cartpole.

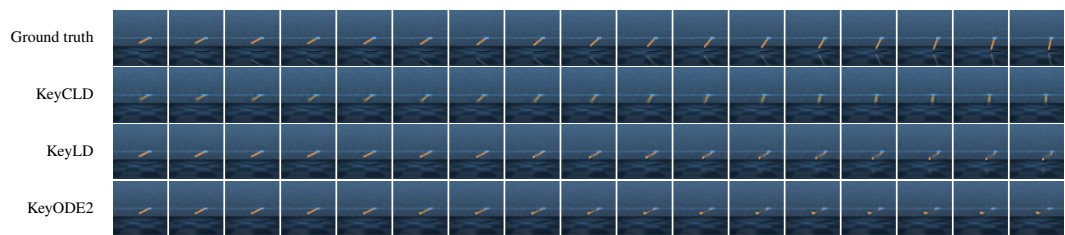

Figure 11: Underactuated cartpole.

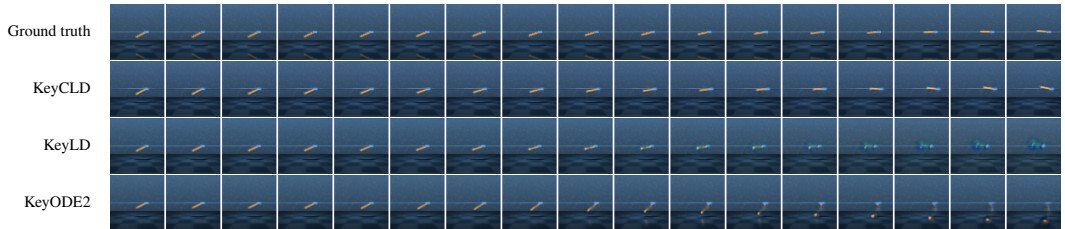

Figure 12: Fully actuated cartpole.

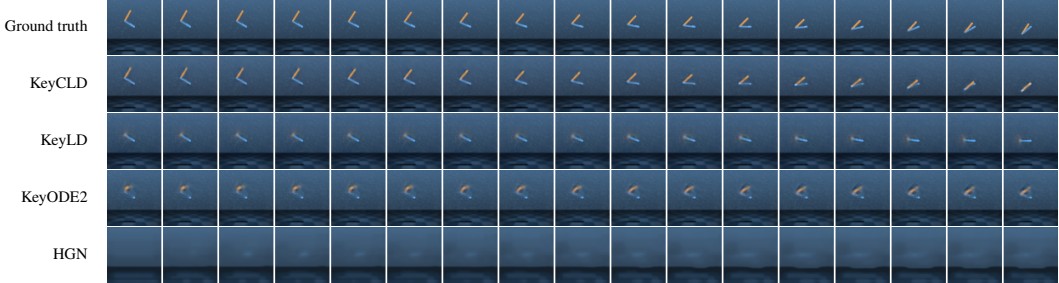

Figure 13: Unactuated acrobot.

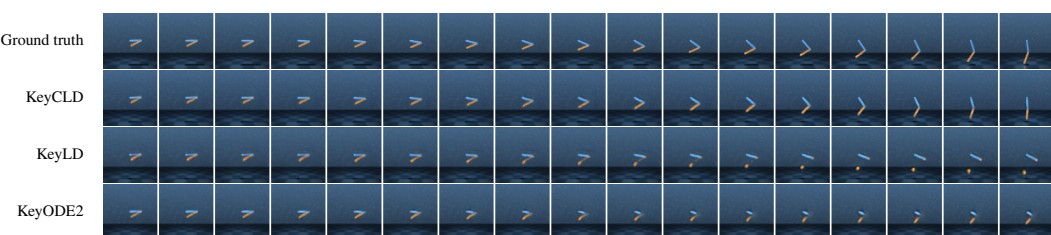

Figure 14: Underactuated acrobot.

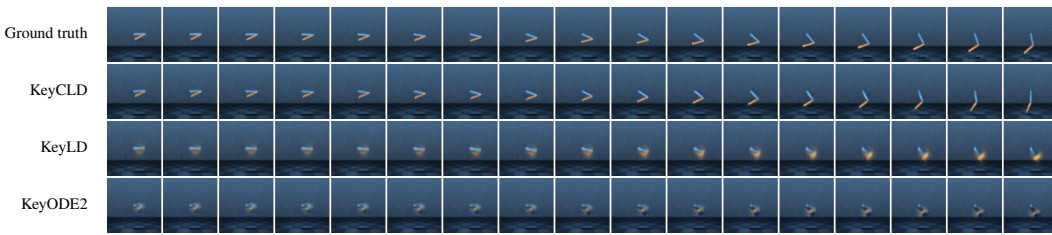

Figure 15: Fully actuated acrobot.

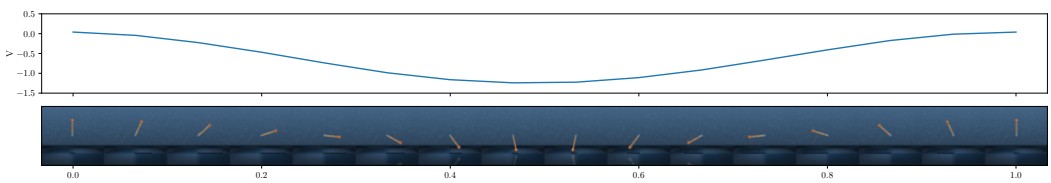

Figure 16: Potential energy of the trained KeyCLD model of the pendulum environment. The pendulum makes a full rotation. As expected, the potential energy follows a smooth sinusoidal path throughout this sequence. The maximum value is reached when the pendulum is upright, and the minimum value is reached when the pendulum is down.

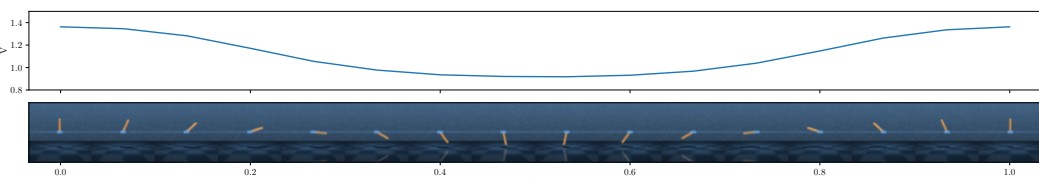

Figure 17: Potential energy of the trained KeyCLD model of the cartpole environment. The position of the cart is fixed, and the pole makes a full rotation. As expected, the potential energy follows a smooth sinusoidal path throughout this sequence.

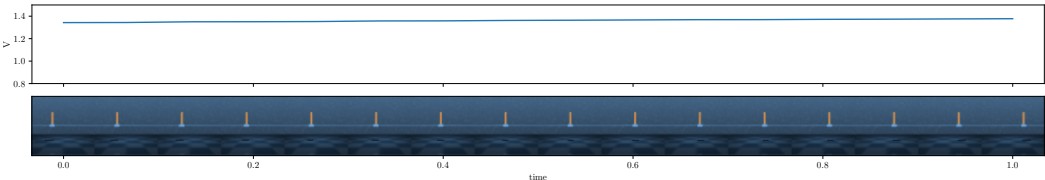

Figure 18: Potential energy of the trained KeyCLD model of the cartpole environment. The pole is fixed and the cart moves from left to right. As expected, the change in potential energy in this sequence is very low (compare to Fig. 17 with the same axis). A horizontal movement has no impact on the gravity potential.

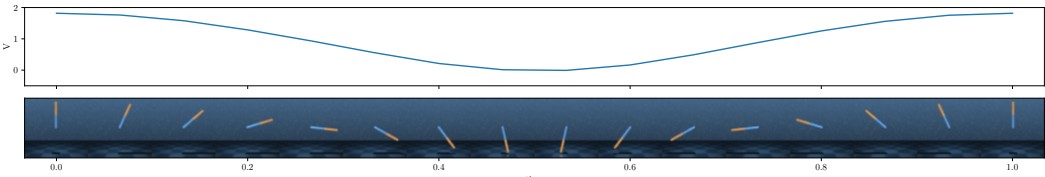

Figure 19: Potential energy of the trained KeyCLD model of the acrobot environment. The first link makes a full rotation, the second link is fixed relative to the first link. As expected, the potential energy follows a smooth sinusoidal path throughout this sequence.

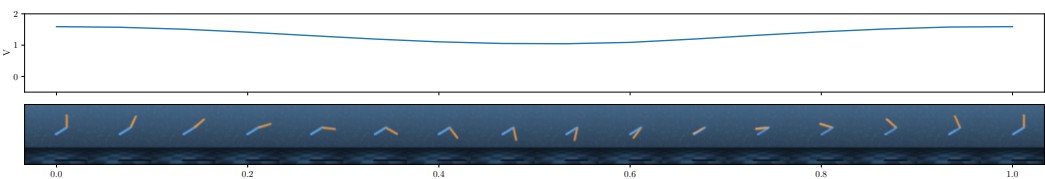

Figure 20: Potential energy of the trained KeyCLD model of the acrobot environment. The first link is fixed and the second link makes a full rotation. Again, the potential energy follows a smooth sinusoidal path throughout this sequence. Please compare with Fig. 19, where both links are moving. Here the potential energy changes less, because the first link is not moving.

