# OpenReview forum: "KeyCLD: Learning Constrained Lagrangian Dynamics in Keypoint Coordinates from Images"
_ICLR.cc/2023/Conference — Submitted to ICLR 2023_

### Official Review · Reviewer_NWGA · 2022-10-21

**Confidence:** 4
**Correctness:** 2
**Technical Novelty And Significance:** 2
**Empirical Novelty And Significance:** 2
**Recommendation:** 5

**Clarity, Quality, Novelty And Reproducibility:**

The paper is not very well written and deserves improvements. The model description is hard to follow. Novelty is very limited, and the proposed method is not situated among previous works. Reproducibility seems possible, but code release would have been appreciated.

**Strength And Weaknesses:**

In my opinion, the novelty of the proposed method is extremely limited. I list below a several papers (sometimes absent from the cited references) comparable to the work presented:

- [1] introduces a very similar framework for unsupervised 2D keypoint detection. The presented model is a simplification in that visual information is no longer extracted from a source image, but simply learned as a fixed latent vector. The model is therefore unlikely to generalize to basic changes in the scene, such as the background color.

- [2] is an extension of [1] presenting a keypoint detection model for control. Their manipulation task is quite similar to those used in paper, although in my opinion more complex. This reference is missing from the paper, and yet seems very relevant.

- [3] offers a different keypoint-based approach in a realistic environment for controlling a 3D robot arm. It seems to me that the method is comparable with this reference (also missing from the citations), and address a much more complex problem.

- [4] and [5] also propose to learn the dynamics of complex systems from the discovery of unsupervised 2D keypoints. However, these two papers use a much more challenging counterfactual context, and address 3D problems.

More generally, the paper is quite difficult to read, section 2 and 3, as well as figures 3 and 4 can in my opinion be re-worked to make understanding easier. Deep understanding of [1], [2] and cited works on Deep Lagrangian models feel absolutely necessary to fully understand the proposition. I list below  several remarks concerning the method:

- The use of strong and demanding prior knowledge on the geometry of the problem seems to be a very strong assumption. Adding its constraints unsurprisingly improves overall performance, as a big part of the dynamic is manually encoded into the network. However, I wonder how precise these constraints need to be. It would have been interesting to show that the use of partial or slightly erroneous constraints made it possible to maintain good performance.

- The authors detail in equation 11 the formula used to estimate the speed of keypoints by finite difference. The authors have chosen to use a forward Euler diagram, in other words, the speed (i) already contains the information of the position at the instant (i+1)! This effect fades when the prediction is made over a longer horizon, but it is a fairly significant design flaw.

- The scenarios chosen to evaluate the model seem terribly simple to me regarding the tasks that SOTA address with this type of keypoints-based model (see references given above). These are simulated 2D problems, with relatively simple dynamics, with very simple visual output (no occlusion, constant background, fixed geometry, etc.)

- It would have been interesting to study in more detail the behavior of detected keypoints. I wonder for example (1) what is the contribution of the BCE on the heatmaps, (2) what is the impact of the increase/reduction of the number of keypoints (3) Is the application of "transformation of the $D_4$ dihedral group" to $z$ useful in practice? It also seems to me that given the constraints on the keypoints, the learned dynamics should be very close to the physical equations, but this is not discussed in the paper.

[1] Jakab, T., Gupta, A., Bilen, H., & Vedaldi, A. (2018). Unsupervised learning of object landmarks through conditional image generation. Advances in neural information processing systems, 31.

[2] Kulkarni, T. D., Gupta, A., Ionescu, C., Borgeaud, S., Reynolds, M., Zisserman, A., & Mnih, V. (2019). Unsupervised learning of object keypoints for perception and control. Advances in neural information processing systems, 32.

[3] Manuelli, L., Li, Y., Florence, P., & Tedrake, R. (2021, October). Keypoints into the Future: Self-Supervised Correspondence in Model-Based Reinforcement Learning. In Conference on Robot Learning (pp. 693-710). PMLR.

[4] Li, Y., Torralba, A., Anandkumar, A., Fox, D., & Garg, A. (2020). Causal discovery in physical systems from videos. Advances in Neural Information Processing Systems, 33, 9180-9192.

[5] Janny, S., Baradel, F., Neverova, N., Nadri, M., Mori, G., & Wolf, C. (2022, April). Filtered-CoPhy: Unsupervised Learning of Counterfactual Physics in Pixel Space. In International Conference on Learning Representation

**Summary Of The Paper:**

This paper proposes to learn the dynamics of simple systems from visual observation. The proposed method is based on the detection of 2D keypoints in the image in an unsupervised way and the learning of the dynamics in its Lagrangian form. Prior knowledge of the structure of the problem is necessary.

**Summary Of The Review:**

Finally, the main novelty of this paper consists in modeling the dynamics in its Lagrangian form. Even if the idea has a certain interest and seems reasonable, I am not convinced by the experiments of the practical advantages of such approach over existing baselines. Furthermore, the simplicity of the proposed tasks, the absence of relevant citations, and of comparisons with the state of the art do not allow me to recommend this work for a presentation at ICLR.

---

> ### Author Response · Authors · 2022-11-10
> **Rebuttal**
>
> Dear reviewer,
>
> Thank you for your review.
>
> [limited novelty] None of the [1-5] papers learn kinetic energy, potential energy and input matrix models. This is of course the main focus of our work, using Lagrangian dynamics.
> This allows accurate, long-term predictions, and very simple energy shaping control (see Appendix 6 for explanation, derivation and results).
>
> [deep understanding of Lagrangian] We’re sorry to hear it is not clear. We did our best to introduce the different concepts sequentially, starting from the general Lagrangian formulation towards the constrained Lagrangian. We also hope including the implementation of equation (6) in JAX in Appendix 2, helps to make the method less abstract and comprehensible.
>
> [prior knowledge] This is a very good question. Only the Jacobian of the constraint function is used. This means that distances between points are not given. Only which points remain at a constant distance from each other. Or in the case of the cartpole that one point stays on a horizontal line. Slightly erroneous constraints are thus not possible, and because of the more discrete search aspect, we hope that learning the constraints from the data will be possible in future work.
>
> [forward Euler] I do not understand why this would be a problem? The whole sequence is of course available during training, so we want to maximally use it. It should be seen as a smoother, rather than a filter approach, to estimate the velocities.
> Of course, when applied for control (see Appendix 6), we do not use information from the future, and do finite differencing with the previous timestep.
> Just to be clear, training is done with an adaptive timestep RK45 ode solver, not with forward Euler.
>
> [simple tasks] As stated before, our goal is to learn Lagrangian dynamics from images, and we clearly perform better than SOTA methods in this domain. In fact, the main focus of our work was to benchmark on more difficult datasets. Earlier methods used far more simple datasets with simple sprites on flat backgrounds.
>
> [more detail] Thank you for these suggestions. I will try to answer them here:
>
> 1. We added BCE loss on the heatmaps to make training more robust. The model would sometimes deteriorate to a state where the heatmap would not highlight a point, but rather a whole region in the image. It would get stuck there during training. The BCE loss makes sure that a point is predicted in the heatmap
>
> 2. The number of keypoints is connected to the definition of the constraint function. We did not increase the number of keypoints as we wanted a parsimonious representation for easier interpretation and control.
>
> 3. In our experiments, it was very useful, especially for the pendulum. The keypoint estimator model was more robust throughout the state space. Before, we had some problems with an offset relative to the tip of the pendulum, that changed throughout the rotation. And training is more robust generally.
>
> 4. [physical equations] Indeed, please see figures 16-20 for insights in the learned potential energy models. Since the model is learned from images, it is not possible to directly compare with known physical equations. It depends on the exact positions of keypoints etc. If you have suggestions for interesting ways to compare this qualitatively or quantitatively, I would be very happy to hear!
>
> [code release] The code is included in the supplementary materials. See the folder ‘code’.
> Installation instructions and instructions to run all reported experiments are in the README.md

---

> > ### Comment · Reviewer_NWGA · 2022-11-22
> > **Answer**
> >
> > I thank the authors for providing answers to my remarks. These help me to better understand the contribution.
> >
> > [limited novelty] Although I understand that your main contribution focuses on learning the lagrangian dynamics from images, I still believe that theses references should be cited in your work (this is not the case for [2], [3] and potentially [4] and [5] with minor impact). For me, the keypoint detector and renderer are not novel, except for the supplementary BCE loss. This is ambiguous in the main paper and should be stated very clearly.
> >
> > [forward Euler] I think there is still something I do not understand. First, thank you for clarify that you did not use future state during control, could you point where this is stated in the paper ? To me, this could also be a problem with the main experiment of the paper, ie. the prediction task. Did you also change the velocity estimation formula for this experiment? If so, where is this specified ?
> >
> > As far as I understand, by using the Lagrangian form you can avoid the dynamics directly exploiting future information bias. Although, I think it should be explained more clearly in the text, especially since I am not confident that the forward euler scheme is essential for the model to learn correctly.
> >
> > Obviously, reviewer U2jo has an impact on my rating, which is reinforced by the fact that my comments, although addressed, did not lead to an update to the paper of any kind. I admit, however, that the paper investigates an interesting problem. Yet, in my opinion, the contributions are rather poorly situated in relation to existing work, and the general presentation of the method and the results are not convincing enough.
> >
> > I updated my rating, but still think this paper is not yet ready for presentation at ICLR. I will carefully follow discussions with other reviewers, and may change my rating accordingly.

---

> > > ### Author Response · Authors · 2022-11-24
> > > **Answer**
> > >
> > > Thank you for your further comments and raising your rating.
> > > [limited novelty] It seems I am too late to be able to change the manuscript anymore. [2], [3] are certainly relevant, and I would be willing to add them.
> > > The main novelty is the connection between constrained Lagrangian dynamics, expressed in Cartesian coordinates, and keypoints. Where keypoints and Cartesian coordinates are one and the same.
> > >
> > > [forward Euler] Since the control algorithm determines the action, and thus the next state, it is quite impossible to use future states for control?
> > > In the caption of Fig. 5 we wrote "50 frames are predicted based on the first three frames of the ground truth sequence to estimate the velocity". The velocity estimation is identical to the training (equation 11). There, i = 2 for the prediction task. The velocity at frame 2 is estimated using frame 1 and frame 3. And starting from frame 2, all the next frames are predicted.

---

### Official Review · Reviewer_U2jo · 2022-10-25

**Confidence:** 5
**Clarity, Quality, Novelty And Reproducibility:** See above
**Correctness:** 3
**Technical Novelty And Significance:** 2
**Empirical Novelty And Significance:** 2
**Recommendation:** 5

**Strength And Weaknesses:**

Dear authors, greeting from your Neurips reviewers. Please keep in mind that the review community for this topic is not soo big. Therefore, there is a high likelihood that you get the same reviewers. This paper deeply frustrates me, as this paper is soo close to the finish line but it seems like you cannot push it over the finish line. You had 2 months and 4 good reviews to adapt the paper. Some of the NEURIPS reviews might be a bit harsh on the baselines but overall fair, positive, and constructive. However, you decided to do very little and resubmit mostly the same paper. Yes, you cut a few equations, added a baseline (that nobody asked for), and reported VPT for no-actuation, under-actuated, and fully actuated. However, you failed to add a qualitative comparison of the prediction. You could easily plot the trajectories in phase space, which would enable a qualitative comparison, and plot the constraint violation of the key points. Another fun idea would have been to infer the constraints from the key points which should be straightforward (when the key points are good) compared to hard-coding the constraints. I still find the approach of learning dynamics from pixels but simultaneously requiring the knowledge of the kinematic chain a weird combination. Looking at the videos in the `extra_predictions` does not increase my confidence in the methods. The time horizons are very short and it often fails horribly for the under-actuated and un-actuated versions.

I would urge the reviewers to go back to the NEURIPS reviews and see how the paper could be improved. To refresh your mind, this is the NEURIPS Meta reviews which provide a good summary and positiveness of the different reviewers. Unfortunately, the meta reviewer was a bit too confident as the work was not `substantially improved` but resubmitted with minor changes.

> While all reviewers agree with the novelty and interestingness of this research direction, in its current form, my assessment is that it would make a good workshop paper proposing a 1st step into the direction of learning Lagrangian dynamics from images. The results are still in early stages, e.g. experiments using the simplest DM_control environments (even requiring modifications from the authors on friction to make it work). As uztn noted, “the system is built from parts that are well-described in the prior work, as described by the authors. The resulting system will not work outside of simple 2D, fully observable systems, and the generations produced are of comparable or lower quality than generations of dynamic systems both in and outside of the physically-inspired dynamics modeling community.”
>
> I also agree with reviewer sEpU that a comparison to video-prediction models (or image-based “world models”) is not outside the scope, since people will want to know where the approach stands compared with traditional methods (even at lower performance). As the reviewer mentioned, experiments which enable the comparison to other methods such as control / video-prediction, which are not necessarily Lagrangian-based or are Lagrangian based but do not learn from images (to answer questions like "how much does the use of the Lagrangian improve the video-prediction/control performance" or "how large is the performance drop by learning from images instead states directly") will be tremendously useful to achieve high impact within the broader ML research community.
>
> I look forward to seeing the authors improve the work, as we all agree that the direction is novel and I’m confident that the work will be substantially improved going forward.


**Summary Of The Paper:**

This paper proposes to combine keypoint extraction from vision with physics-inspired deep networks to learn accurate dynamics models from images. The keypoint extractor extracts cartesian coordinates on the individual bodies. The consecutive dynamics model utilizes the key points to learn the constrained dynamics of these bodies. In addition, the proposed algorithm enforces Lagrangian dynamics on each body. In contrast to prior work, this paper uses a keypoint extractor while the other approaches utilized a VAE. Within the experiments, the authors show that the proposed algorithm can learn a good dynamics model.

**Summary Of The Review:**

While I do not object accepting the paper, I don't think that the quality of this paper is ready for publication at ICLR. If the other reviewers, want the paper to be published, I am fine with that.

---

> ### Author Response · Authors · 2022-11-10
> **Rebuttal**
>
> Dear reviewer,
>
> Thank you for reviewing our work, a second time.
> We are sorry to hear that you think we only did very little since the last submission, because that is certainly not the case. This resubmission is not at all a simple redaction of the manuscript.
> We focused on making the method stronger, and adding more experiments to allow a more insightful discussion about the benefits of a Lagrangian prior, as was requested by previous reviews.
> We did many experiments to improve the dynamics loss function, which is now completely different from the NeurIPS submission and led to better results. We added augmentation with the D4 dihedral group, which, especially for the pendulum, made the results much more robust.
> Multiple reviewers had questions about underactuated and unactuated systems, and unactuated systems allow direct comparison with HGN.
> That is why we added these experiments, so that was certainly asked for.
> Qualitative comparison of the predictions are included in the appendix and supplementary materials.
> Trajectories of the keypoints are plotted in the supplementary materials, in the ‘long_term_predictions’ movies (subfigure c). We could not add more figures due to page limitations (in favor of e.g. architecture details of the convnet models, as was requested by multiple reviewers).
>
> Inferring the constraint function from the data is indeed a very interesting idea.
> However, it is not as simple as it sounds. It is precisely the interplay between the modeling with constrained Lagrangian dynamics and Cartesian keypoints, which enables the model to find useful keypoints. In other words, the given constraint function helps the model to find good representations.
> We made the choice to make it harder on the vision aspect, and easier on the dynamics aspect. We already iterated on many different ideas and worked for more than a year to reach these results. It is considerably harder to learn Lagrangian dynamics from more realistic images such as dm_control, than from the very simplistic datasets prior work uses.

---

### Official Review · Reviewer_ooc4 · 2022-10-26

**Confidence:** 5
**Correctness:** 3
**Technical Novelty And Significance:** 3
**Empirical Novelty And Significance:** 2
**Recommendation:** 6

**Clarity, Quality, Novelty And Reproducibility:**

*Quality:* For the most part, the submission seems to be technically sound. There are some errors in the text and some of the claims/assertions are not supported (see questions/comments)

*Clarity:* For the most part, the submission is clearly written and well organized (see questions/comments for concerns/suggestions)

*Originality:* The approach appears new and is well situated in the context of previous work.


**Strength And Weaknesses:**

*Strengths:* The authors address the challenging problem of designing physics informed deep neural networks with interpretable representations. The proposed method outperforms other methods on the proposed dataset and can learn to control underactuated systems.

*Weaknesses:* I would have liked to see a comparison between the proposed approach and Lag-caVAE on the dataset given in Lag-caVAE. I would also be interested in a comparison in the number of parameters between models.

*Notes:*
- “and introduced energy-based control” – I think this phrasing may be confusing to some readers, since energy based control was not a contribution of Zhong et al.
- “This enables more data efficient models, at the cost of providing extra knowledge about the system in the form of a constraint function.” – was this shown?
- The authors claim that the use of a fully convolutional model is an advantage of the proposed approach over Zhong & Leonard, 2020 because of the translation equivariance. How does translation equivariance benefit the model when most of the experiments vary under rotation?
- The authors note that additional information must be given to the renderer to compensate for translation equivariance, but an equivariant representation shifts under the relevant transformation, why would additional information be necessary?

*Questions:*
- What do the authors mean by interpretable coordinates in Table 1? For example, by construction, the latent representation learned in HGN is interpretable as the generalized position and generalized momentum; however, the authors do not mark it as having interpretable coordinates.

*Possible typos:*
- “wether they are” → whether they are
- “changes in function of the state” → changes as a function of the state
- “ are now in function of” →  are now a function of
- “The Hamiltonian expresses the total energy of the system” – I think the citation used here is inappropriate; consider using a well known classical mechanics textbook instead.
- “Lagrangian mechanics because it is more general” – I think the citation used here is inappropriate; consider using a well known classical mechanics textbook instead.


**Summary Of The Paper:**

The authors introduce a method to learn Lagrangian dynamics from images. In line with previous work, the authors simplify the learning problem by: (1) representing the Lagrangian as the difference in kinetic and potential energy (both learned), and (2) using Cartesian coordinates to represent the positions of the system components. The main contribution of this work is the approach for identifying the positions of the system components. The authors do this by extracting keypoints from image inputs. The authors demonstrate the utility of their approach on the dm_control dataset.

**Summary Of The Review:**

The method is well motivated and demonstrates strong performance relative to baseline models. Some of the claims/assertions are not well supported.

---

> ### Author Response · Authors · 2022-11-10
> **Rebuttal**
>
> Dear reviewer,
>
> Thank you for your thoughtful review.
>
> [comparison with Lag-caVAE on respective dataset]
> Thank you for the suggestion. Working towards more realistic images is one of the main points of our work, that is why we report benchmarks on dm_control.
> I understand evaluating on the simplistic dataset is still relevant, but will not be feasible during rebuttal.
>
> [number of parameters] We will include this in the manuscript.
>
> [more data-efficient] This is shown in Finzi et al. (2020). We did not provide specific experiments.
>
> [advantage of translation equivariance] Also for e.g. the acrobot, the second part can have the same orientation and be positioned at different locations.
> Furthermore, since we augment with rotations of 90 degrees and flips (D4 dihedral group), this effect is multiplied.
>
> [renderer additional information] To reconstruct the background image, some positional encoding is necessary, in our method this is in the form of the learned seed tensors. Also for moving parts and reflections, this positional information helps to reconstruct, since appearance depends on the location of the keypoints in the image.
>
> [interpretable coordinates] We mean coordinates with a physical meaning, real positions in the world. As opposed to latent, abstract coordinates.
>
> Thank you for the other remarks, suggestions and typo’s. We will adapt the manuscript.

---

### Decision · Program_Chairs · 2023-01-20

**Decision:**

Reject

**Justification For Why Not Higher Score:**

Too simple setup (observed 2d systems and known constraints).

**Justification For Why Not Lower Score:**

The paper makes a small steps in an important direction.

**Metareview: Summary, Strengths And Weaknesses:**

The papers proposes an algorithm for learning keypoint representations of images with the assumption that the dynamics of the keypoints is constrained Lagrangian dynamics. The model is shown to produce good predictions on three environments (Pendulum, Cartpole and Acrobot).

The reviewers acknowledge the general direction of using physics-motivated inductive biases when learning representations. This work makes a small step in this direction which is appreciated by the reviewers. However, the setup considered in the paper is rather simple: 1) the model is applicable only to fully observed 2d systems and 2) the constraints are known (not inferred). Especially the second assumption seems unrealistic.

Given the simplicity of the considered setup, the reviewers would like to see more thorough evaluation of the model and qualitative assessment of the experimental results. For example: how good are the extracted key points, how much do they violate the constraints, qualitative assessment of the importance of the constraints, what is the main challenge of inferring the constraints from data. Such analysis would provide more insight on the model's strengths and limitations. For these reasons, the reviewers do not recommend acceptance of the paper but they encourage the authors to improve the paper and re-submit.

**Summary Of Ac-Reviewer Meeting:**

The reviewers acknowledge the general direction of using physics-motivated inductive biases when learning representations. This work makes a small step in this direction which is appreciated by the reviewers. However, the setup considered in the paper is rather simple: 1) the model is applicable only to fully observed 2d systems and 2) the constraints are known (not inferred). Especially the second assumption seems unrealistic. Given the simplicity of the considered setup, the reviewers would like to see more thorough evaluation of the model and qualitative assessment of the experimental results. For example: how good are the extracted key points, how much do they violate the constraints, qualitative assessment of the importance of the constraints, what is the main challenge of inferring the constraints from data. Such analysis would provide more insight on the model's strengths and limitations. For these reasons, the reviewers do not recommend acceptance of the paper but they encourage the authors to improve the paper and re-submit.